# Association between serum retinol and overall and cause-specific mortality in a 30-year prospective cohort study

Jiaqi Huang [1,2✉], Stephanie J. Weinstein [2], Kai Yu[2], Satu Männistö [3] & Demetrius Albanes[2✉]

How retinol as a clinical indicator of vitamin A status is related to long-term mortality is unknown. Here we report the results of a prospective analysis examining associations between serum retinol and risk of overall and cause-specific mortality. During a 30-year cohort follow-up, 23,797 deaths were identified among 29,104 men. Participants with higher serum retinol experienced significantly lower overall, CVD, heart disease, and respiratory disease mortality compared to men with the lowest retinol concentrations, reflecting 17–32% lower mortality risk ($P_{trend} < 0.0001$). The retinol-overall mortality association is similar across subgroups of smoking intensity, alcohol consumption, body mass index, trial supplementation, serum alpha-tocopherol and beta-carotene concentrations, and follow-up time. Mediation analysis indicated that <3% of the effects of smoking duration and diabetes mellitus on mortality were mediated through retinol concentration. These findings indicate higher serum retinol is associated with lower overall mortality, including death from cardiovascular, heart, and respiratory diseases.

[1] National Clinical Research Center for Metabolic Diseases, Key Laboratory of Diabetes Immunology, Ministry of Education, and Department of Metabolism and Endocrinology, The Second Xiangya Hospital of Central South University, Changsha, Hunan 410011, China. [2] Division of Cancer Epidemiology and Genetics, National Cancer Institute, NIH, Department of Health and Human Services, Bethesda, MD, USA. [3] Department of Public Health Solutions, Finnish Institute for Health and Welfare, Helsinki, Finland. ✉email: JIAQI.HUANG@CSU.EDU.CN; DAA@NIH.GOV

Vitamin A is a class of essential, fat-soluble compounds of which retinol is the major form in circulation and considered the biochemical indicator of human vitamin A status[1]. Animal products are the primary food sources of retinol, including liver, butter, milk, cheese, and egg. Retinol and related retinoids act as ligands for cellular retinoic acid and retinoid X receptors (RARs and RXRs) through which they modulate transcription of genes regulating a wide-range of physiological processes including embryonic development, post-natal growth, vision, epithelial differentiation and maintenance, and immunity[1–3]. The deficiency of vitamin A can lead to xerophthalmia and blindness, infections and inflammation, chronic diseases, and even death. Despite the well-established clinical importance of vitamin A sufficiency for growth and development, physiological homeostasis, and overall human health, a comprehensive assessment of the long-term relationship between vitamin A status and overall and cause-specific mortality is lacking.

Population-based studies have demonstrated inconsistent associations between circulating retinol and risk of cancer, cardiovascular disease (CVD), and death[4–10]. A meta-analysis of four studies revealed no material association between circulating vitamin A (retinol) and all-cause mortality[11]. However, most of the previous studies were of limited sample size and the number of events (i.e., ranging from 62 to 720 deaths)[4–6,9,10], controlled for confounding factors inconsistently (e.g., only controlling for age and sex[6]), and had relatively low power to examine cause-specific mortality, effect modification by other factors[4–9], and dose-response associations[4–6,9].

In order to more robustly evaluate the vitamin A-mortality relationship, we conducted a prospective analysis of a large cohort of men with serum retinol measured at study entry and over three decades of subsequent observation with nearly 24,000 deaths from heart disease, stroke, other CVD, cancer, respiratory disease, diabetes mellitus, and other causes. Dose-response and population subgroups of lifestyle and other selected factors were also examined. Here, we show statistically significant inverse associations between higher serum retinol and lower risk of overall and cause-specific mortality, including death from cardiovascular, heart, and respiratory diseases. The inverse retinol-overall mortality association is similar across several cohort subgroups, including smoking intensity, alcohol consumption, body mass index, trial supplementation, serum alpha-tocopherol, and beta-carotene concentrations, and duration of follow-up.

## Results

**Baseline characteristics.** The median baseline serum retinol concentration in the cohort was 576 μg/L (interquartile range = 500 μg/L to 662 μg/L), and the mean retinol value in the fifth quintile was 84% higher than that in the first quintile. According to the WHO criteria, the present study had only two participants identified with severe vitamin A deficiency (≤98 μg/L), and 21 participants identified with subclinical vitamin A deficiency (98 to ≤196 μg/L). When compared with men in the lowest quintile category, men in higher quintiles of retinol were slightly younger, had higher blood pressure, BMI, and educational level, and were more likely to have a history of CVD, use vitamin supplements and be more physically active (Table 1). Serum retinol was also positively associated with serum total and HDL cholesterol and alcohol consumption (as expected)[12–14], and inversely associated with serum beta-carotene. With regard to the magnitude of the correlation, serum retinol was negligibly or weakly correlated with these factors (all Pearson correlation coefficients, $-0.12 < r \leq 0.22$)[15].

**Serum retinol and overall and cause-specific mortality.** During 31 years of follow-up (median 18 years), there were 514,289 person-years of accumulated observation, with 23,797 deaths, including 9,869 deaths from CVD (8,064 heart disease and 1,764 stroke), 7,695 deaths from cancer, 2,161 deaths from respiratory disease, 119 deaths from diabetes, 1,255 deaths from injuries and accidents, and 2,698 deaths from other causes. The age-adjusted analysis showed that, compared with those in the lowest quintile, men in the higher quintiles of serum retinol experienced lower mortality from all-causes combined, cancer, and respiratory disease, reflecting 10−38% reductions (fifth versus first quintile, HRs [95% CIs]: 0.90 [0.87, 0.94], 0.90 [0.83, 0.96], 0.62 [0.54, 0.71], respectively; all $P_{trend} \leq 0.0013$; Table 2). The multivariate-adjusted models for mortality from all-causes combined, CVD, heart disease, respiratory disease and other causes, indicated 17−32% lower mortality among men in the highest quintile (fifth versus first quintile, HRs [95% CIs]: 0.83 [0.80, 0.87], 0.81 [0.76, 0.86], 0.79 [0.74, 0.85], 0.68 [0.59, 0.78], 0.83 [0.74, 0.94], respectively; all $P_{trend} < 0.001$; overall mortality per one SD HR, 0.95 [0.94, 0.96]). The findings of absolute risk difference for the associations remained qualitatively similar when truncating the follow-up observation periods at 10- or 20-years. The adjusted 30-year one SD ARDs (95% CIs) were −1.19% (−1.54%, −0.87%), −0.80% (−1.17%, −0.43%), −0.83% (−1.20%, −0.44%), −0.87% (−1.25%, −0.53%), and −0.75% (−1.18%, −0.30%), for all-causes combined, CVD, heart disease, respiratory disease and other causes, respectively (Table 3).

The restricted 4-knot cubic spline linear regression of the serum retinol-mortality association substantiated a non-linear dose-response, with all-cause and cause-specific mortality (mortality from CVD and heart disease) increasing among men with serum retinol concentrations below 483 μg/L (cutoff reference value of the lowest quintile) (Fig. 1). Increasing serum retinol was related to decreasing overall, CVD and heart disease mortality, with greatest risk reduction apparent for concentrations near 660 μg/L. The multivariable fractional polynomials analysis, after adjustment for confounding factors, showed similar patterns of the dose-response associations between serum retinol and overall mortality as well as CVD and heart disease mortality (P values < 0.0001 for multivariable fractional polynomials compared with linear, Supplemental Fig. 1). By contrast, serum retinol appeared to be linearly associated with respiratory disease mortality (on the log scale of the hazard ratio) (restricted cubic splines model, P nonlinear = 0.04, Fig. 1; multivariable fractional polynomials model, P nonlinear > 0.05, Supplemental Fig. 1). Kaplan−Meier survival plots show that compared with those in the higher quintiles, men in the lowest category of serum retinol experienced excess cumulative overall mortality and mortality from CVD, heart disease and respiratory disease (all log-rank P values < 0.0001; Fig. 2).

**Stratified analyses.** Stratified analyses revealed no significant differences in the retinol-overall mortality association across subgroups of smoking intensity, alcohol consumption, BMI, trial supplementation, serum alpha-tocopherol, and beta-carotene concentrations, and duration of follow-up (Fig. 3). There was, however, a significant interaction with both age and serum total cholesterol where the inverse association appeared stronger among men who were older than 54 years and those with low total cholesterol concentrations ($P_{interaction} < 0.005$, respectively). Findings were similar across subgroups for CVD, heart disease, and respiratory disease mortality with no significant differences observed (Supplemental Tables 1–3).

**Table 1 Baseline characteristics by quintile of serum retinol [a] (all reported *P*-values are two-sided at type I error rate of 0.05).**

| | Quintile of serum retinol | | | | |
| --- | --- | --- | --- | --- | --- |
| | Quintile 1 (n = 5,815) | Quintile 2 (n = 5,874) | Quintile 3 (n = 5,741) | Quintile 4 (n = 5,840) | Quintile 5 (n = 5,834) |
| Serum retinol (µg/L) | 424 (52.5) | 516 (18.7) | 577 (17.1) | 642 (22.3) | 781 (93.4) |
| Age (y) | 58.1 (5.2) | 57.5 (5.1) | 57.2 (5.0) | 56.8 (5.0) | 56.4 (4.9) |
| Cigarettes/d | 20 (8) | 20 (8) | 20 (9) | 21 (9) | 21 (9) |
| Duration of smoking (y) | 37.3 (8.3) | 36.6 (8.3) | 35.9 (8.3) | 35.2 (8.6) | 34.7 (8.6) |
| Systolic blood pressure (mm Hg) | 140 (20) | 141 (19) | 141 (19) | 142 (19) | 145 (20) |
| Diastolic blood pressure (mm Hg) | 86 (11) | 87 (11) | 87 (11) | 88 (11) | 90 (11) |
| Serum total cholesterol (mmol/L) | 5.8 (1.1) | 6.1 (1.1) | 6.3 (1.1) | 6.4 (1.1) | 6.6 (1.2) |
| Serum HDL cholesterol (mmol/L) | 1.17 (0.30) | 1.18 (0.31) | 1.19 (0.31) | 1.20 (0.32) | 1.24 (0.35) |
| Serum beta-carotene (µg/L) | 210 (168) | 218 (180) | 221 (188) | 215 (178) | 194 (204) |
| BMI (kg/m$^2$) | 25.5 (4.0) | 26.1 (3.8) | 26.3 (3.8) | 26.5 (3.7) | 26.9 (3.7) |
| Education (%, >elementary school) | 18.2 | 18.2 | 20.5 | 22.6 | 25.7 |
| Physically active (%) | 17.7 | 21.2 | 21.4 | 22.5 | 21.3 |
| History of CVD (%)[b] | 40.3 | 40.3 | 39.3 | 41.2 | 46.2 |
| History of diabetes mellitus (%) | 5.0 | 4.3 | 3.8 | 3.9 | 4.2 |
| Vitamin A supplement use (%) | 8.6 | 9.1 | 10.2 | 10.9 | 13.0 |
| Vitamin E supplement use (%) | 8.0 | 9.2 | 10.1 | 10.8 | 12.7 |
| Daily dietary intake | | | | | |
| Energy (kcal) | 2,666 (765) | 2,697 (759) | 2,698 (739) | 2,711 (753) | 2,672 (751) |
| Fat (triacylglycerol, g) | 106 (37) | 107 (37) | 107 (36) | 106 (36) | 103 (36) |
| Alcohol (g ethanol) | 13.5 (19.1) | 14.5 (18.6) | 16.8 (20.2) | 19.8 (21.9) | 25.4 (25.3) |
| Vitamin A (mg) | 1.78 (1.05) | 1.83 (1.07) | 1.87 (1.10) | 1.91 (1.07) | 1.94 (1.12) |
| Vitamin E (mg) | 12.2 (6.1) | 12.1 (5.8) | 12.0 (5.7) | 12.1 (5.6) | 11.8 (5.5) |
| Fruit (g) | 127 (101) | 130 (101) | 129 (103) | 130 (99) | 129 (106) |
| Vegetables (g) | 107 (70) | 110 (69) | 114 (70) | 118 (72) | 118 (72) |
| Red meat (g) | 70 (34) | 71 (35) | 72 (34) | 72 (33) | 72 (34) |

[a] Values are means with standard deviation unless otherwise indicated. ANOVA test for continuous variables and Chi-square test for categorical variables. All *P* value < 0.0001, with exception of fruit intake (*P* value > 0.05).
[b] CVD, cardiovascular disease; includes a history of deep vein thrombosis, superficial venous thrombosis, lung infarction or embolus, hypertension, arterial obstruction, stroke, heart arrhythmia, enlarged heart, valvular heart disease, myocardial infarction, coronary heart disease, and heart failure.

**Sensitivity analyses**. The inverse serum retinol associations with mortality from CVD, heart disease, and respiratory disease were accentuated after excluding the 12,488 men with a baseline history of CVD or diabetes mellitus (mortality from CVD, heart disease, and respiratory disease in the multivariable-adjusted model, fifth versus first quintile: HRs = 0.72, 0.68 and 0.60, respectively; all $P_{trend} < 0.0001$). By contrast, the retinol-overall mortality association remaining unchanged (HR = 0.81, $P_{trend} < 0.0001$; Supplemental Table 4). The findings were similar after excluding the first 5 years of follow-up (Supplemental Table 5) and when used serum retinol concentration data from the third follow-up year (fifth versus first quintile, HR [95% CIs] for overall mortality: 0.85 (0.81, 0.89), $P_{trend} < 0.0001$; Supplemental Table 6).

**Mediation analyses**. As reported, traditional risk factors including obesity, intensity of smoking, duration of smoking, diabetes mellitus, and alcohol drinking were statistically significantly associated with increased risk of overall mortality, whereas light or moderate physical activity was inversely associated risk of mortality in the cohort (Table 4). Our results from the mediation analysis suggest that 2.2% (1.5−3.2) of the smoking duration association with mortality, as well as 1.2% (0.5−3.1) of the diabetes mellitus association, were mediated through serum retinol concentration.

**Discussion**
In this large prospective serological analysis involving 29,104 cohort participants and 23,797 deaths over three decades, we found significant inverse associations between serum vitamin A (retinol, subsequently referred as "vitamin A") and mortality from overall, CVD, heart disease and respiratory disease after adjusting for several important risk factors, with subjects in the highest serum quintile experiencing 17–32% lower mortality when compared with those in the lowest quintile category (5–12% per one SD increment). The beneficial overall mortality association was evident across all population subgroups with the exception of being stronger among older men and those with lower serum total cholesterol. The effect sizes of these observed risk estimates were relatively small, however.

The present investigation is the largest to examine the association of vitamin A biochemical concentrations with all-cause and cause-specific mortality, and our data suggests mortality declines with increasing retinol concentration, with a risk nadir for overall and CVD mortality in older men with serum values of 600–700 µg/L, and excess mortality among men with retinol below 500 µg/L (n = 7,321, which included two and 21 participants that exhibited severe [≤98 µg/L] or subclinical [98 to ≤196 µg/L] vitamin A deficiency, respectively). Humans cannot synthesize vitamin A and derive this essential nutrient from animal product-based retinyl esters and plant-based pro-vitamin A carotenoids (e.g., alpha- and beta-carotene)[16,17]. For example, the primary dietary sources for dietary vitamin A in this population (as mean percentage of total daily dietary vitamin A) were liver (33.9%), butter (16.4%), egg (9.2%), milk (8.6%), cheese (3.5%) and other food items combined (28.4%). Preformed retinol is absorbed from the intestine, esterified, and transported in chylomicrons to the liver. For healthy, well-nourished individuals, it is estimated that 60–95% of vitamin A is stored in the liver which plays a central role in its metabolism and homeostasis through hydrolysis of stored retinyl esters, complexing of retinol with retinol-binding protein (RBP), and release of this "holo-RBP" into systemic circulation for uptake and use by other

**Table 2 Hazard ratios (HRs) and 95% CI for overall and cause-specific mortality by quintile of serum retinol [a](we used Cox proportional hazard regression models to estimate HRs and 95% CIs. All reported P-values are 2-sided at type I error rate of 0.05. The Bonferroni corrected P trend value is also presented).**

| Causes of mortality | Serum retinol (ug/L) | | | | | P for trend | Bonferroni corrected P for trend | Per one SD (130 µg/L) |
|---|---|---|---|---|---|---|---|---|
| | Quintile 1 | Quintile 2 | Quintile 3 | Quintile 4 | Quintile 5 | | | |
| **Overall** | | | | | | | | |
| Deaths (n) | 5,036 | 4,825 | 4,640 | 4,588 | 4,708 | | | |
| Death rate[b] | 53.78 | 46.75 | 44.62 | 42.67 | 44.45 | | | |
| Age-adjusted HR (95% CI)[c] | 1.00 | 0.87 (0.84, 0.91) | 0.85 (0.81, 0.88) | 0.82 (0.79, 0.86) | 0.90 (0.87, 0.94) | <0.0001 | <0.0001 | 0.97 (0.96, 0.98) |
| Multivariate HR (95% CI)[d] | 1.00 | 0.87 (0.83, 0.90) | 0.84 (0.80, 0.87) | 0.80 (0.77, 0.83) | 0.83 (0.80, 0.87) | <0.0001 | <0.0001 | 0.95 (0.94, 0.96) |
| **CVD** | | | | | | | | |
| Deaths (n) | 2,051 | 1,919 | 1,974 | 1,901 | 2,024 | | | |
| Death rate[b] | 21.90 | 18.59 | 18.98 | 17.68 | 19.11 | | | |
| Age-adjusted HR (95% CI)[c] | 1.00 | 0.86 (0.81, 0.91) | 0.89 (0.84, 0.95) | 0.85 (0.80, 0.90) | 0.97 (0.91, 1.03) | 0.50 | 1.00 | 1.01 (0.99, 1.03) |
| Multivariate HR (95% CI)[d] | 1.00 | 0.82 (0.77, 0.88) | 0.85 (0.79, 0.90) | 0.77 (0.72, 0.82) | 0.81 (0.76, 0.86) | <0.0001 | <0.0001 | 0.95 (0.93, 0.97) |
| **Heart disease** | | | | | | | | |
| Deaths (n) | 1,674 | 1,568 | 1,625 | 1,549 | 1,648 | | | |
| Death rate[b] | 17.88 | 15.19 | 15.63 | 14.41 | 15.56 | | | |
| Age-adjusted HR (95% CI)[c] | 1.00 | 0.86 (0.80, 0.92) | 0.90 (0.84,0.97) | 0.85 (0.79, 0.91) | 0.96 (0.90, 1.03) | 0.46 | 1.00 | 1.01 (0.99, 1.03) |
| Multivariate HR (95% CI)[d] | 1.00 | 0.82 (0.76, 0.88) | 0.85 (0.79, 0.91) | 0.76 (0.71, 0.82) | 0.79 (0.74, 0.85) | <0.0001 | <0.0001 | 0.95 (0.92, 0.97) |
| **Stroke** | | | | | | | | |
| Deaths (n) | 369 | 347 | 340 | 343 | 365 | | | |
| Death rate[b] | 3.94 | 3.36 | 3.27 | 3.19 | 3.45 | | | |
| Age-adjusted HR (95% CI)[c] | 1.00 | 0.86 (0.74, 0.99) | 0.85 (0.74, 0.99) | 0.84 (0.73, 0.98) | 0.97 (0.84, 1.13) | 0.89 | 1.00 | 1.01 (0.96, 1.06) |
| Multivariate HR (95% CI)[d] | 1.00 | 0.85 (0.74, 0.99) | 0.84 (0.73, 0.98) | 0.81 (0.69, 0.94) | 0.86 (0.74, 1.01) | 0.084 | 0.76 | 0.98 (0.93, 1.03) |
| **Cancer** | | | | | | | | |
| Deaths (n) | 1,583 | 1,607 | 1,506 | 1,520 | 1,479 | | | |
| Death rate[b] | 16.90 | 15.57 | 14.48 | 14.14 | 13.96 | | | |
| Age-adjusted HR (95% CI)[c] | 1.00 | 0.93 (0.87, 1.00) | 0.88 (0.82, 0.94) | 0.87 (0.81, 0.94) | 0.90 (0.83, 0.96) | 0.0013 | 0.012 | 0.96 (0.94, 0.99) |
| Multivariate HR (95% CI)[d] | 1.00 | 0.95 (0.88, 1.01) | 0.90 (0.84, 0.96) | 0.89 (0.83, 0.96) | 0.92 (0.85, 0.99) | 0.016 | 0.14 | 0.97 (0.94, 0.99) |
| **Respiratory disease** | | | | | | | | |
| Deaths (n) | 554 | 465 | 402 | 399 | 341 | | | |
| Death rate[b] | 5.92 | 4.51 | 3.87 | 3.71 | 3.22 | | | |
| Age-adjusted HR (95% CI)[c] | 1.00 | 0.77 (0.68, 0.87) | 0.67 (0.59, 0.76) | 0.66 (0.58, 0.76) | 0.62 (0.54, 0.71) | <0.0001 | <0.0001 | 0.83 (0.79, 0.87) |
| Multivariate HR (95% CI)[d] | 1.00 | 0.82 (0.73, 0.93) | 0.74 (0.65, 0.84) | 0.74 (0.65, 0.85) | 0.68 (0.59, 0.78) | <0.0001 | <0.0001 | 0.88 (0.84, 0.93) |

**Table 2 (continued)**

| Causes of mortality | Serum retinol (ug/L) | | | | | P for trend | Bonferroni corrected P for trend | Per one SD (130 µg/L) |
|---|---|---|---|---|---|---|---|---|
| | Quintile 1 | Quintile 2 | Quintile 3 | Quintile 4 | Quintile 5 | | | |
| **Diabetes mellitus** | | | | | | | | |
| Deaths (n) | 24 | 17 | 21 | 30 | 27 | | | |
| Death rate[b] | 0.26 | 0.16 | 0.20 | 0.28 | 0.25 | | | |
| Age-adjusted HR (95% CI)[c] | 1.00 | 0.63 (0.34, 1.18) | 0.78 (0.44, 1.41) | 1.09 (0.64, 1.87) | 1.04 (0.60, 1.81) | 0.36 | 1.00 | 1.08 (0.90, 1.30) |
| Multivariate HR (95% CI)[d] | 1.00 | 0.64 (0.34, 1.19) | 0.79 (0.44, 1.43) | 1.00 (0.57, 1.73) | 0.88 (0.49, 1.56) | 0.85 | 1.00 | 1.02 (0.85, 1.23) |
| **Injuries and accidents** | | | | | | | | |
| Deaths (n) | 256 | 262 | 214 | 244 | 279 | | | |
| Death rate[b] | 2.73 | 2.54 | 2.06 | 2.27 | 2.63 | | | |
| Age-adjusted HR (95% CI)[c] | 1.00 | 0.93 (0.78, 1.11) | 0.76 (0.63, 0.91) | 0.84 (0.71, 1.01) | 0.98 (0.83, 1.17) | 0.83 | 1.00 | 0.99 (0.94, 1.05) |
| Multivariate HR (95% CI)[d] | 1.00 | 0.94 (0.79, 1.12) | 0.76 (0.63, 0.92) | 0.83 (0.69, 1.00) | 0.91 (0.76, 1.09) | 0.26 | 1.00 | 0.99 (0.93, 1.05) |
| **Other causes** | | | | | | | | |
| Deaths (n) | 568 | 555 | 523 | 494 | 558 | | | |
| Death rate[b] | 6.07 | 5.38 | 5.03 | 4.59 | 5.27 | | | |
| Age-adjusted HR (95% CI)[c] | 1.00 | 0.87 (0.77, 0.98) | 0.81 (0.72, 0.91) | 0.75 (0.66, 0.85) | 0.91 (0.81, 1.02) | 0.053 | 0.48 | 0.96 (0.92, 1.00) |
| Multivariate HR (95% CI)[d] | 1.00 | 0.87 (0.77, 0.98) | 0.80 (0.71, 0.90) | 0.73 (0.64, 0.82) | 0.83 (0.74, 0.94) | 0.0011 | 0.01 | 0.92 (0.88, 0.96) |

Abbreviations: *BMI* body mass index, *CI* confidence interval, *CVD* cardiovascular disease, *HDL* high–density lipoprotein, *HR* hazard ratios, *SD* standard deviation
[a]Cause of death was missing for 47 subjects.
[b]Crude death rate per 1000 person-years.
[c]Adjusted for age. P value for trend: based on statistical significance of the coefficient of the quintile variable (median value within each quintile).
[d]Adjusted for age, BMI, serum total and serum HDL cholesterol, cigarettes smoked per day, years of smoking, alcohol intake, intervention assignment, systolic and diastolic blood pressure, history of CVD, and history of diabetes.

**Table 3 Absolute risk differences for the association between serum retinol and overall and cause-specific mortality.**

| Cause of mortality | Absolute risk difference, % (95% CI)[a] | | |
| --- | --- | --- | --- |
| | 10-year risk difference | 20-year risk difference | 30-year risk difference |
| Overall | −0.52 (−0.68, −0.38) | −1.01 (−1.33, −0.75) | −1.19 (−1.54, −0.87) |
| CVD | −0.25 (−0.37, −0.14) | −0.57 (−0.84, −0.31) | −0.80 (−1.17, −0.43) |
| Heart disease | −0.25 (−0.37, −0.14) | −0.57 (−0.83, −0.31) | −0.83 (−1.20, −0.44) |
| Stroke | −0.023 (−0.072, 0.030) | −0.071 (−0.22, 0.094) | −0.13 (−0.41, 0.17) |
| Cancer | −0.15 (−0.25, −0.038) | −0.38 (−0.65, −0.10) | −0.56 (−0.94, −0.15) |
| Respiratory disease | −0.092 (−0.13, −0.055) | −0.46 (−0.66, −0.28) | −0.87 (−1.25, −0.53) |
| Diabetes mellitus | $-6.6 \times 10^{-4}$ ($-7.9 \times 10^{-3}$, $7.9 \times 10^{-3}$) | $5.2 \times 10^{-3}$ (−0.056, 0.060) | $1.0 \times 10^{-3}$ (−0.10, 0.11) |
| Injuries and accidents | −0.011 (−0.071, 0.056) | −0.023 (−0.14, 0.11) | −0.033 (−0.21, 0.17) |
| Other causes | −0.077 (−0.12, −0.030) | −0.31 (−0.49, −0.12) | −0.75 (−1.18, −0.30) |

[a]Adjusted absolute risk difference was calculated as a difference of one SD serum retinol over the follow-up of 10, 20, 30 years, respectively. The 95% CIs were estimated from 500 bootstrap samples. Models were adjusted for age, BMI, serum total and serum HDL cholesterol, cigarettes smoked per day, years of smoking, alcohol intake, intervention assignment, systolic and diastolic blood pressure, history of CVD, and history of diabetes.
Abbreviations: *BMI* body mass index, *CI* confidence interval, *CVD* cardiovascular disease, *HDL* high-density lipoprotein.

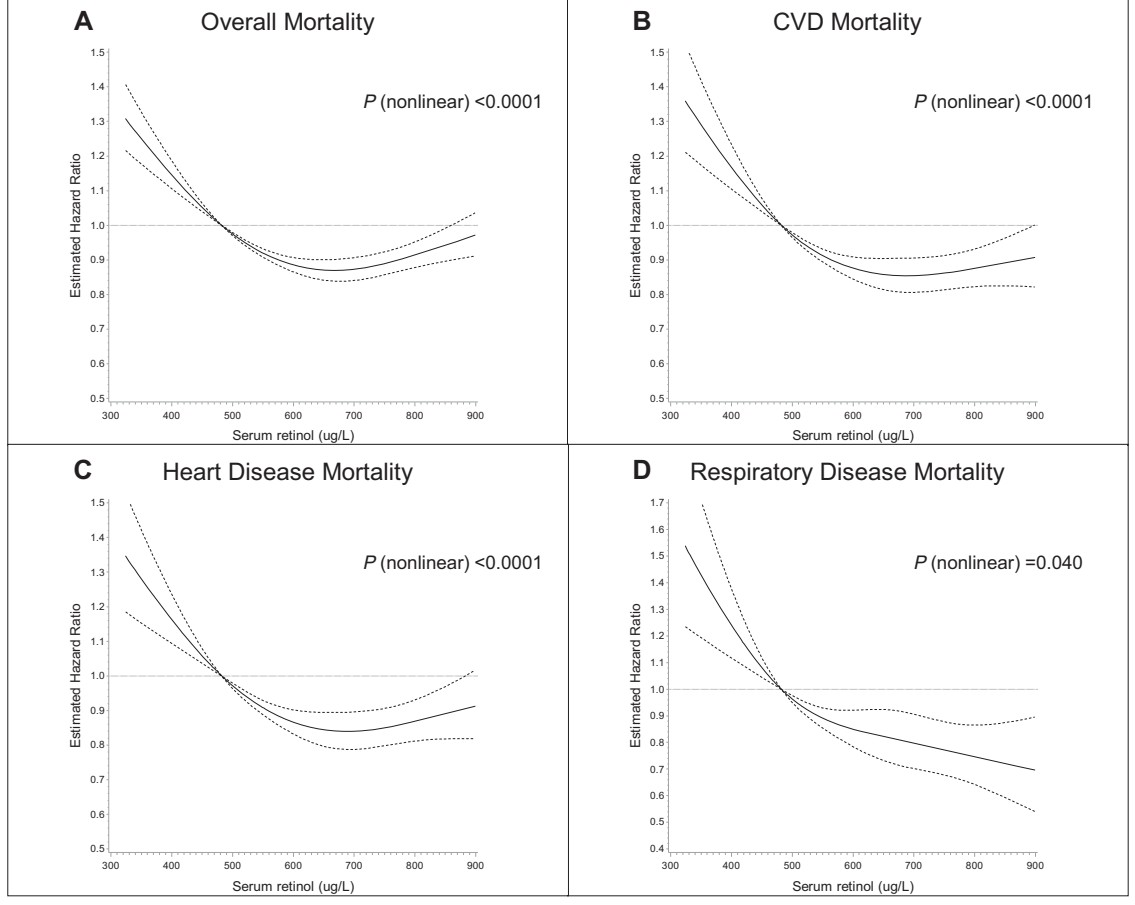

**Fig. 1 Cubic spline regression for the association between serum retinol and overall and cause-specific death in the ATBC Study.** The reference value (483 µg/L; hazard ratio = 1) corresponds to the cutoff value of the first quintile category of serum retinol concentration. **A** Overall mortality. **B** Cardiovascular disease (CVD) mortality. **C** Heart disease mortality. **D** Respiratory disease mortality. The solid line represents the hazard ratio (HR) for mortality and serum retinol with a 4-knot spline (knots were selected at the 5th, 25th, 75th, and 95th percentiles of the serum retinol); dashed lines suggested the 95% confidence intervals. The total number of participants: 29,104. Event number of overall-, CVD-, heart disease-, respiratory disease death is 23,797, 9,869, 8,064, and 2,161, respectively. Multivariate-adjusted models adjusted for age, BMI, serum total and serum HDL cholesterol, cigarettes smoked per day, years of smoking, alcohol intake, intervention assignment, systolic and diastolic blood pressure, history of CVD, and history of diabetes. ATBC Alpha-Tocopherol, Beta-Carotene Cancer Prevention; BMI body mass index; CVD cardiovascular disease; HDL high-density lipoprotein.

organs[16–19]. Previous studies raised the possibility that both insufficient and excess vitamin A status may be associated with increased risk of adverse health outcomes[8,20], including as a result of direct toxic properties of hypervitaminosis A[17]. We did not observe excess mortality among men with high serum retinol, however, possibly because of a limited range of concentrations in our study population. The threshold of toxic concentrations of retinol, as well as whether there is a positive association of

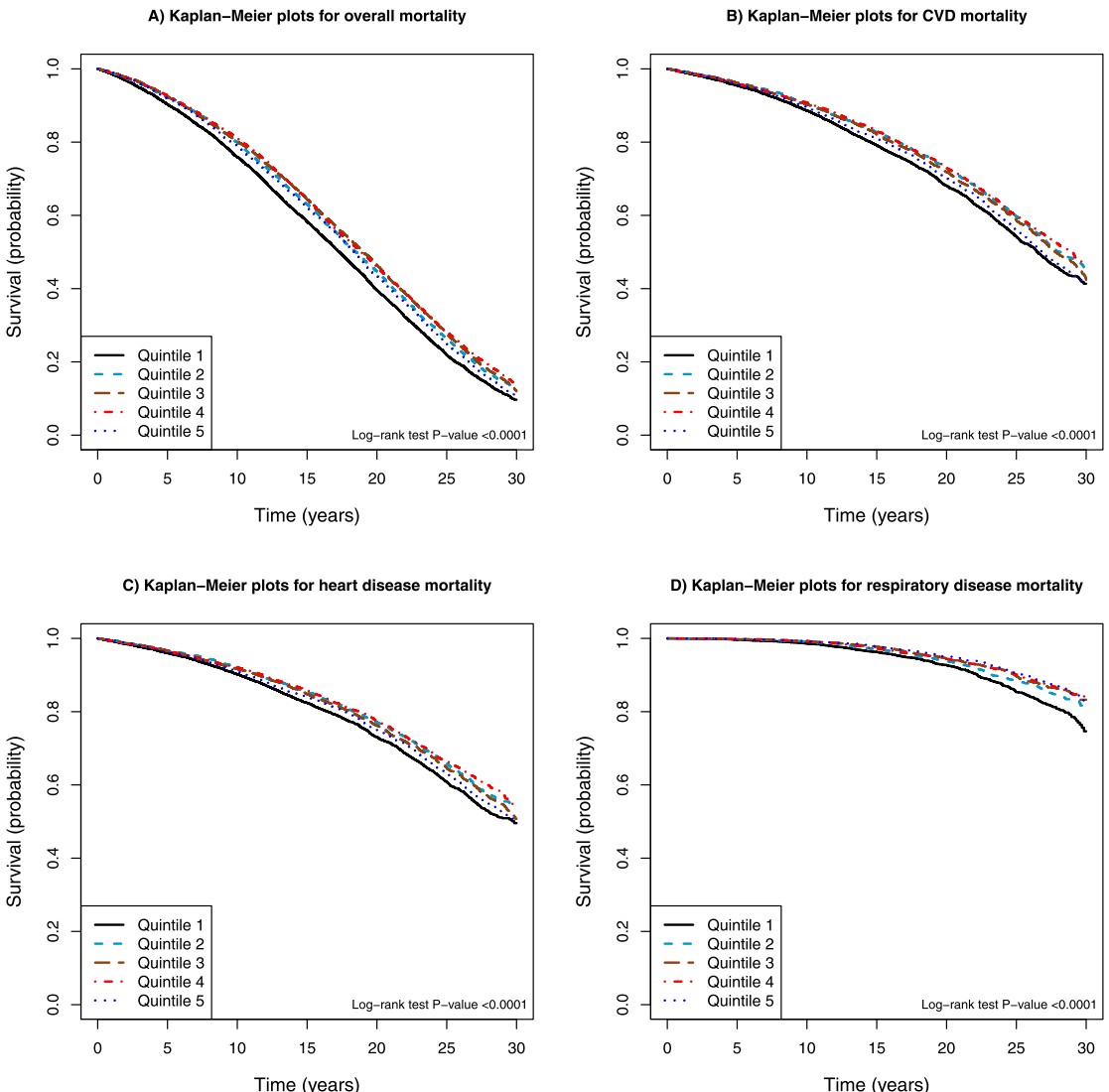

**Fig. 2 Kaplan–Meier plots of overall and cause-specific mortality according to quintile categories of serum retinol in 29,104 participants in the ATBC Study. A** Overall mortality. **B** Cardiovascular disease (CVD) mortality. **C** Heart disease mortality. **D** Respiratory disease mortality. ATBC Alpha-Tocopherol, Beta-Carotene Cancer Prevention; CVD cardiovascular disease.

mortality risk with excess concentrations of serum retinol, can be evaluated in further studies of populations with available data of excessive serum retinol values. Given the fact that vitamin A is metabolized and stored in the liver[17], a positive association between liver consumption and serum retinol is biologically plausible. However, along with previous findings[12–14,21,22], our data showed a negligible association between serum retinol and dietary liver consumption (Pearson correlation coefficient = 0.05, $P < 0.0001$), and a weak association with alcohol consumption (Pearson correlation coefficient = 0.20, $P < 0.0001$), with negligible and weak correlations being defined as correlation coefficients that range from 0.00 to 0.10 and 0.10 to 0.39, respectively[15]. Additionally, our data showed that the inverse association between serum retinol and risk of overall mortality was not changed by additional adjustment for liver consumption (Q5 versus Q1: HR = 0.84, 95% CI: 0.80, 0.87, $P$ for trend < 0.0001). This suggests that a vitamin A-enriched diet was not sufficient to modify the associations appreciably.

Findings from prior studies regarding vitamin A status and overall mortality have been conflicting, with several investigations showing no associations, dose-response or otherwise[4–8]. The NHANES III Study (The Third National Health and Nutrition

Examination Survey) of 16,008 participants with 4,225 deaths showed a U-shaped association between serum retinol and total mortality such that individuals with concentrations in either the quintile 3 or 4 category (but not in quintile 5) experienced 18% reduced total mortality compared with those in the lowest quintile (HRs 95% CIs = 0.82 [0.68, 0.98] and 0.82 [0.69, 0.97], respectively; $P_{trend} = 0.93$)[7]. Using the same NHANES III Study with 6,069 participants aged 50 years or older, a U-shaped overall mortality association was found with HRs of 2.9 and 1.2 for participants with vitamin A status that was deficient (<30 μg/dL) or excessive (>80 μg/dL) when compared to those with normal retinol concentrations (30–80 μg/dL)[8], respectively. In a cohort of 379 renal transplant recipients with a median of 1,739 days follow-up, serum retinol concentration was related to a decreased risk of all-cause mortality[9]. For established mortality risk factors, our mediation analysis suggests that only a small, <3% role is played by serum retinol for the association of overall mortality with the duration of smoking and diabetes mellitus. Whether retinol mediates portions of other risk factor-mortality associations (e.g., lipid impairment) can be investigated in future studies.

Retinol biochemical status and cause-specific mortality, particularly CVD mortality, has been only sporadically studied, with

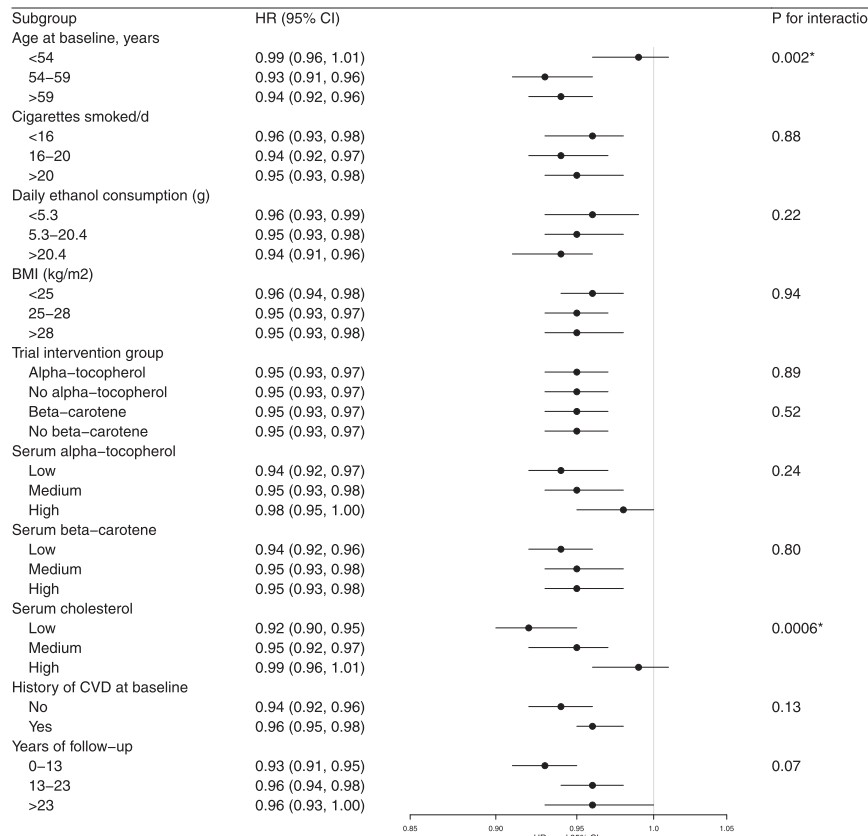

**Fig. 3 Hazard ratios for overall mortality by serum retinol, stratified by selected factors.** Hazard ratios of overall mortality were for one SD increment of serum retinol concentration. We used Cox proportional hazard regression models to estimate HRs and 95% CIs. Models were adjusted for age, BMI, serum total and serum HDL cholesterol, cigarettes smoked per day, years of smoking, alcohol intake, intervention assignment, systolic and diastolic blood pressure, history of CVD, and history of diabetes. P value for interaction: according to the likelihood test to assess the statistical significance of the cross-product term entered into the Cox proportional hazard regression model. Abbreviations: BMI body mass index; CI confidence interval; CVD cardiovascular disease; HDL high-density lipoprotein; HR hazard ratios. *P value for interaction achieved the Bonferroni corrected threshold: 0.05/10 = 0.005.

**Table 4 Mediating influence of serum retinol on the associations between major risk factors and overall mortality.**

| Mortality risk factor | Multivariate HR (95% CI)[a] | Proportion (%) of effect due to mediation through retinol | P value for mediation |
|---|---|---|---|
| Obesity | 1.21 (1.13, 1.30) | None | — |
| Intensity of smoking[b] | 1.28 (1.19, 1.37) | None | — |
| Duration of smoking[b] | 1.30 (1.21, 1.41) | 2.2 (1.−3.2) | <0.0001 |
| Diabetes mellitus | 1.87 (1.72, 1.03) | 1.2 (0.5−3.1) | 0.016 |
| Alcohol drinking[b] | 1.08 (1.01, 1.15) | None | — |
| Physical activity | 0.81 (0.76, 0.85) | None | None |

[a]Cox proportional hazard regression model, adjusted for age, obesity (no: BMI < 30; yes: BMI > = 30), BMI (continuous), years of smoking (quintile), cigarettes smoked per day (quintile), history of diabetes (no or yes, self-reported at baseline), alcohol intake (quintile), physical activity (light or moderate physical activity: no or yes), serum total and HDL cholesterol (continuous), intervention assignment, systolic and diastolic blood pressure (continuous), and history of CVD (no or yes, self-reported at baseline).
[b]Reported HR: quintile 5 (Q5) versus Q1 of major risk factor from the Cox regression model.
Abbreviations: *BMI* body mass index, *CI* confidence interval, *CVD* cardiovascular disease, *HDL* high-density lipoprotein, *HR* hazard ratio.

conflicting findings and several null associations reported[4–7]. By contrast, serum retinoic acid was inversely associated with overall and CVD mortality among 1,499 patients with coronary artery disease, with those in the highest serum quartile experiencing 35 and 40% lower mortality, respectively, as compared with those in the lowest quartile category[10]. In 1,530 acute ischemic stroke patients, lower retinoic acid concentrations were related to higher risk of overall and CVD mortality, independent of other risk factors[23], results that were essentially consistent with the present findings. Case-control studies have also documented inverse associations of plasma retinol with risk of CVD including coronary heart disease[13], stroke[12], and CVD mortality[24]. Such protective vitamin A-CVD mortality associations are supported by several plausible biological mechanisms. Retinol, its derivate retinoids, and their receptors exhibit potent inhibition of the atherosclerotic disease through cholesterol efflux from macrophages, enhanced reendothelialization, and reduced proinflammatory response. Cholesterol efflux capacity has been characterized as reverse cholesterol transport and shown prospectively to be inversely related to CVD events including mortality[25–27]. 9-*Cis*-retinoic acid also suppresses lipoprotein-derived cholesterol accumulation in macrophages and inhibits foam cell formation[28]

and the development of atherosclerosis[29]. Retinoids also exhibit anti-inflammatory, immune-modulatory, anti-angiogenic and anti-thrombotic properties, and they enhance vasodilatory biochemical synthesis, suppress vasoconstriction, reduce injury-reactive intimal thickening, and enhance nitric oxide pathway-related endothelial function[30–36]. In addition, the primary vitamin A carrier in circulation, retinol-binding protein 4 (RBP4), has been related to cardiometabolic and inflammatory markers, and higher RBP4 concentrations are associated with metabolic syndrome and cardiovascular disease[37–40].

In contrast to CVD, there are relatively few studies of circulating vitamin A and respiratory disease mortality. The British National Diet and Nutrition Survey study of 1054 participants observed a non-significant inverse association for respiratory disease mortality with an age- and sex-adjusted 12% lower risk per SD increment in plasma vitamin A[6]. Previous case-control studies showed significantly increased risk of both chronic obstructive pulmonary disease (COPD)[41] and nontuberculous mycobacterial pulmonary disease[42] for persons with lower plasma vitamin A, and a cross-sectional study of 50 COPD patients and controls reported essentially the same[43]. Whether vitamin A inhibition of inflammatory responses and suppression of host respiratory infections can account for the respiratory mortality associations will require further study. The present finding of a weak and non-significant retinol-cancer mortality association after excluding the first 5 years of follow-up, indicating some potential reverse causality, is consistent with the lack of associations reported in previous studies[5–7,44].

The major strength of our study is its basis in a large prospective cohort with complete follow-up up to 31 years and identification of cause-specific mortality via linkage with Finnish national registries. Retinol serological status was determined independently through a high-quality isocratic high-performance liquid chromatography platform for more than 29,000 men, which provided an accurate and objective measurement of retinol biochemical concentrations. Additionally, the large sample size enabled us to examine associations for the less common causes of mortality and test whether other factors such as age, alcohol consumption, or smoking intensity modified the vitamin A-mortality associations. Our findings are also subject to some limitations that should be noted. The cohort was a relatively homogenous population of male Finnish smokers which limits generalizability of our findings to other populations. There was only a single assessment of baseline serum retinol, and the physiological status of some participants may have changed during the long follow-up period. Other biomarkers relevant to vitamin A status, including retinol-binding protein, retinoic acid receptor expression, and C-reactive protein (CRP, a marker of inflammation, that the inflammation status may affect retinol homeostasis and serum retinol concentrations), that would have afforded a deeper evaluation of the mortality associations, were not available for the cohort. Underestimated mortality associations towards the null may result from biases attributed to non-differential misclassification such as measurement error. However, we found the baseline and year-3 retinol concentrations were highly correlated (Spearman correlation coefficient = 0.69, $P$-value < 0.0001), indicating its relative stability during the post-randomization period. Despite having controlled for a large number of potential confounding factors, the possibility of findings being subject to residual confounding from other unmeasured factors cannot be ruled out.

In conclusion, our data indicate that higher vitamin A status is associated with lower overall and cause-specific mortality in men. The associations were based on over 30 years of follow-up and remained unchanged after adjusting for potential confounding risk factors and were largely stable across a range of population subgroups. The study provides evidence supporting the importance of vitamin A biochemical sufficiency for long-term health and longevity. Reexamination of these associations in populations that include women, nonsmokers, and other races/ethnicities is warranted.

## Methods

**Study cohort**. The present analysis is nested within the Alpha-Tocopherol, Beta-Carotene Cancer Prevention (ATBC) Study cohort, details of which have been documented (ClinicalTrials.gov Identifier: NCT00342992)[45]. This randomized, double-blind, placebo-controlled trial evaluated whether alpha-tocopherol or beta-carotene could decrease cancer incidence (particularly lung cancer). The study enrolled 29,133 male smokers, aged 50–69 years from 1985 to 1988 in southwestern Finland. The participants were assigned to one of four supplementation groups—alpha-tocopherol (50 mg/day), beta-carotene (20 mg/day), both vitamins, or placebo—for 5–8 years until the intervention ended on April 30, 1993. At the baseline visit, questionnaires obtained data for demographics, medical and smoking histories, and lifestyle characteristics, including a modified food frequency questionnaire and color picture booklet that queried consumption frequency and portion sizes for 203 food items and 73 mixed dishes for the previous 12 months. Height, weight, and blood pressure were measured by professional nurses, and pre-supplementation fasting blood samples were processed to serum and stored at −70 °C. The ATBC Study was conducted in accordance with the requirement by the Declaration of Helsinki. All participants provided written informed consent, and the study was reviewed and approved by the Institutional Review Boards at the US National Cancer Institute and the Finnish National Public Health Institute.

**Laboratory measurements**. Serum concentrations of retinol, alpha-tocopherol, and beta-carotene were measured for 29,104 of the participants (99.9%) using reversed-phase high-performance liquid chromatography, and the coefficient of variation for serum retinol was 3.1%[45,46].

**Prospective follow-up and causes of death**. Cohort participant person-time was calculated from randomization date during 1985–1988 until date of death or the end of follow-up (December 31, 2015), whichever occurred first. Specific causes of death were obtained through linkage with the Finnish Death Registry of Statistics Finland (Supplemental Fig. 2). Codes from three revisions of *International Classification of Diseases* (the eighth, ninth, and tenth revisions: the ICD-8, ICD-9, ICD-10) were used to define underlying causes of death: CVD mortality (ICD-8 = 390–458; ICD-9 = 390–459; ICD-10 = I00–I99), heart disease mortality (ICD-8 and ICD-9 = 390–398, 401–404, 410–429, and 440–448; ICD10 = I00–I13, I20–I51, and I70–I78), stroke mortality (ICD-8 and ICD-9 = 430–438; ICD10 = I60–I69), cancer mortality (ICD-8 and ICD-9 = 140–239; ICD10, C00-D48), respiratory disease mortality (including influenza, pneumonia, COPD, and other related conditions; ICD-8 = 470–474, 480–486, 490–493, and 518; ICD-9 = 480–487 and 490–496; ICD-10 = J10–J18 and J40–J47), diabetes mortality (ICD-8 and ICD-9 = 250; ICD-10 = E10–E14), injury and accident mortality (ICD-8 = 800–978; ICD-9 = E800–E978; ICD-10 = U01–U03, V01–X59, X60–X84, X85–Y09, Y35, Y85–Y86, Y87.0, Y87.1, and Y89.0), and other causes of mortality combined (including 47 subjects with missing cause of death).

**Statistical analysis**. Cox proportional hazard regression models with attained age as the time metric were used to estimate hazard ratios (HRs) and two-sided 95% confidence intervals (CIs) to examine the association between serum retinol (quintiles and per one SD) and mortality. The proportional hazard assumption was not violated and was tested using model interaction terms of the dichotomized indicator of follow-up years and serum retinol quintiles, with a likelihood ratio test comparing models with and without the interaction terms. A set of a priori covariables were chosen for the risk models based on the existing knowledge and research regarding potential confounding of the serum retinol-mortality risk associations. In the multivariable analysis, we adjusted for age at baseline (continuous), body mass index (BMI, continuous), serum total and serum high-density lipoprotein (HDL) cholesterol (continuous), cigarettes smoked per day (continuous), years of smoking (continuous), alcohol intake (categorical), intervention assignment (alpha-tocopherol supplementation: yes or no; beta-carotene supplementation: yes or no), systolic and diastolic blood pressure (continuous), history of CVD (yes or no), and history of diabetes (yes or no). For each HR from the primary Cox regression model, the adjusted absolute risk difference (ARD) was calculated for each one SD increment of serum retinol for observation periods of 10, 20, and 30 years, with the 95% CIs for ARD being computed based on 500 bootstrap samples. In terms of missing values for any individual covariate (<5%), we generated a missing value indicator in the models.

Restricted cubic splines with four knots were applied to flexibly model the possible nonlinear association between serum retinol levels (continuous) and overall and cause-specific mortality, with the selected change-points being at the 5th, 25th, 75th, 95th percentiles of retinol concentration. Likelihood ratio tests compared differences between the linear term model and the model with both linear and cubic spline terms, with a $P$-value of < 0.0056 indicating nonlinearity

[nine tests]. We further used multivariable fractional polynomials to model the potential nonlinear association between serum retinol and overall and cause-specific mortality (by using R [version 3.5.2] Package "mfp"). Kaplan–Meier survival plots and log-rank tests assessed survival differences across serum retinol quintiles both for all participants and (separately) men without a baseline history of CVD or diabetes mellitus.

In order to evaluate potential interactions and the robustness of our findings, stratified analyses were constructed based on a priori categories of baseline age (<54, 54–<59, or ≥59 years), daily cigarettes (<16, 16–20, or ≥20 cigarettes per day), alcohol consumption (grams of ethanol per day, <5.3, 5.3–<20.4, or ≥20.4, BMI ( <25, 25–<28, or ≥28 kg/m$^2$), intervention group (alpha-tocopherol or no alpha-tocopherol, beta-carotene or no beta-carotene), serum alpha-tocopherol (low: <10.4 mg/L, medium: 10.4–<12.8 mg/L, or high: ≥12.8 mg/L), serum beta-carotene (low: <130 µg/L, medium: 130–<223 µg/L, or high: ≥223 µg/L), serum total cholesterol (low: 5.7 mmol/l, medium: 5.7–<6.6 mmol/l, or high: ≥6.6 mmol/l), self-reported CVD history (no, or yes), and years of follow-up (<13, 13–<23, or ≥23 years). Potential effect modification was examined through likelihood ratio tests that entered interaction terms for serum retinol (one SD) and each stratified factor into the multivariate-adjusted model and compared regression models with and without the interaction terms. Secondary analyses were restricted to participants without a history of CVD or diabetes mellitus ($n = 12,488$ men excluded). In a lag analysis, we further excluded the first 5 years of follow-up to minimize the potential bias due to reverse causation ($n = 2,727$ men excluded). In a sensitivity analysis, we examined serum retinol concentrations measured from blood collected in year three in relation to subsequent overall and cause-specific mortality ($n = 22,312$ men included). Baseline and year 3 serum retinol were highly correlated (Spearman correlation coefficient = 0.69).

We also performed a mediation analysis to quantify what portion of the association between overall mortality and known risk factors (including obesity, intensity of smoking, duration of smoking, diabetes mellitus, alcohol drinking and physical activity) was mediated through serum retinol concentration. Using the above-mentioned Cox proportional regression model, we estimated the HRs and 95% CIs, and adjusted for age, obesity (no: BMI < 30; yes: BMI ≥ 30), BMI (continuous), years of smoking (quintile), cigarettes per day (quintile), history of diabetes (no or yes, self-reported at baseline), alcohol consumption (quintile), physical activity (light or moderate physical activity: no or yes), serum total and HDL cholesterol (continuous), intervention assignment, systolic and diastolic blood pressure (continuous), and history of CVD (no or yes, self-reported at baseline). The traditional risk factors examined were mutually adjusted. We used the %MEDIATE SAS macro to estimate the mediation proportion and 95% CIs of serum retinol for the traditional risk factor-mortality associations[47].

All statistical tests were performed using SAS software (version 9.4; SAS Institute Inc, Cary, NC) and the R statistical language (version 3.5.2; Vienna, Austria). All reported P-values are two-sided at type I error rate of 0.05. The Bonferroni correction threshold was applied to control for multiple tests (0.05/9 = 0.0056 for primary and secondary analyses [nine tests], and 0.05/10 = 0.005 for ten effect modification tests in subgroups).

**Reporting summary**. Further information on research design is available in the Nature Research Reporting Summary linked to this article.

## Data availability
Because of previously enacted EU Data Protection Regulation (GDPR) privacy rules and an existing data use agreement between Finland and the U.S. National Cancer Institute, the ATBC Study data, and materials described in the manuscript may not be made indiscriminately publicly available for the purposes of reproducing the findings. Please contact the ATBC Study Principal Investigators for appropriate specific data requests (https://atbcstudy.cancer.gov/). Data requests are subject to approval by the study review committees. For example, the review committee will check if the data requests containing information that could compromise research participant privacy or participant informed consent. The response time to data requests may be affected during the COVID-19 pandemic.

## Code availability
The analytical methods for the current study are available from the corresponding author upon appropriate request. Data analysis packages using R and SAS software are listed in the Reporting summary.

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

## Acknowledgements

We appreciate all participants in the ATBC Study cohort for their contribution to this research. The ATBC Study is supported by the Intramural Research Program of the U.S. National Cancer Institute, National Institutes of Health. The funder of this study had no role in the design and conduct of the study; collection, management, analysis, and interpretation of the data; preparation, review, or approval of the manuscript; and the decision to submit the manuscript for publication.

## Author contributions

J.H. analyzed and drafted the manuscript. J.H. and D.A. conceived and designed the study. K.Y. checked the accuracy of data analysis. J.H., S.J.W., K.Y., S.M., and D.A. interpreted the results, critically revised the manuscript, and approved the final version of the manuscript. The corresponding author attests that all listed authors meet authorship criteria and that no others meeting the criteria have been omitted. J.H. and D.A. are the study guarantors and accept full responsibility for the work and/or the conduct of the study, had access to the data, and controlled the decision to publish.

## Competing interests

All author declares no competing interests.
