## [Peer Review File · Nature Communications]

Reviewers' Comments:

Reviewer #1:

Remarks to the Author:

This is a review on the biostatistical methods used, not on the general relevance of the paper.

Overall I feel that the analysis has been done carefully. The following remarks may be helpful for revision

- The functional form of the main variable of interest has been investigated using splines. This is fine, another option would have been fractional polynomials. I would have preferred this, however both methods have their pro's and con's. I suspect both methods would have given very similar shapes of the dose-response. An additional analysis e.g. in another supplement would strengthen the analysis.
- There are some in my view unexpected correlations between the baseline covariables, as seen in table 1. This has been mentioned (l 53-55) however is not further discussed. Since for some disease groups there are major differences between the age-adjusted and the multivariable model (CVD, heart disease), further investigations of interaction effects could be useful and interesting.
- The selection procedure for the covariables in the multivariable model is not mentioned. I presume they have been selected *a priori* and no modelling procedure such as backward selection or a more advanced method was applied. Please discuss.
- I find the notation "1-SD" a bit strange (better: "one SD" ?) and I wonder if it is not more illustrative to report the estimated ARD for a fixed value, say 100 µg/l.
- the Kaplan-Meier estimates are illustrative and should not be hidden in the supplement, if possible.

Reviewer #2:

Remarks to the Author:

J Huang et al. present a large prospective epidemiological study on the association between serum retinol concentration and the risk of all-cause and cause-specific death among participants from the ATBC study followed for up to 30y. This is an interesting study about a vitamin which health impact has been neglected for too long, while it could be explained by its highly regulated metabolism (well-known homeostasis, which may induce a very small variability and then no significant results and publication bias). The study has indisputable strengths, including the original topic, the large sample size, the prospective design.

It is generally acknowledged that the circulating retinol concentration is highly regulated, to ensure a secure healthy context. Therefore, exhibiting very low levels of circulating retinol would reflect a potential liver storage issue, while excess of retinol levels would also be accompanied by adverse health outcomes. It's surprising not to have discuss these points here, and not to have define an upper limit not to surpass. However, results did not support such hypothesis.

Abstract

Authors should privilege the most interesting results, also being enough informative. However, here there is no number describing higher risk of death (how many?), no CI 95%. This is important since the sample size may help finding statistically significant results, with low clinical relevance. What are the subgroups? What are the number of cause-specific deaths?

Introduction

This is a short and well-written introduction, which could be more informative about limitations of previous studies (it is mentioned that sample size are limited, such as follow-up durations, confounding factors (which ones?)) and results from diverse sub-sample groups (which ones?) should be underlined.

Methods:

This section is well-written and comprehensively described.

It remains some minor points that should be addressed:

- Why duration and intensity of smoking were both considered as adjustment variables? It appears these variables could be co-linear and may be not both useful in the same model
- Several stratified analyses have been performed, while it should better justify.
- The risk of death after 10, 20 and 30y of follow-up has been reported. Why? Did the authors justify why they considered these thresholds?

-

Results

- Table 1. Please include at the top of each column the pertinent data. Either the median, the mean, or the range of serum retinol concentrations.
- Bonferroni: are all results presented after correction for multiple comparisons ?

Discussion:

- The homeostasis of serum retinol concentrations should be acknowledged as a limit. This means that the small variation of the serum retinol concentration between participants of this sample might rather reflect some particular dietary habits. There are some references about the circulating retinol concentration, which appears associated with dietary intakes (offal such as liver). Adding some data about the dietary intakes of participants would have improve the interpretation of data.
- A single retinol assessment is available; some relevant markers of the vitamin A status (RBP, retinoic acid, retinoic acid receptor expression) would have reinforced the results, while it's easily understandable that such measures were not available for a so large sample.

Overall, it remains minor issues that should be addressed to improve this paper.

Catherine Féart

Reviewer Comments:

Reviewer #1 (Remarks to the Author):

This is a review on the biostatistical methods used, not on the general relevance of the paper. Overall I feel that the analysis has been done carefully. The following remarks may be helpful for revision

Response 1: Thank you for the overall evaluation. We have addressed each of the Reviewer's comments below and have modified the manuscript accordingly.

- The functional form of the main variable of interest has been investigated using splines. This is fine, another option would have been fractional polynomials. I would have preferred this, however both methods have their pro's and con's. I suspect both methods would have given very similar shapes of the dose-response. An additional analysis e.g. in another supplement would strengthen the analysis.

Response 2: Thank you. The Fractional polynomial models utilize generalized transformations to calculate the associations between covariates and the outcome of interest, and typically have either one or two polynomial terms. One of the benefits of applying the fractional polynomial approach is allowing for the evaluation of non-linear associations. According to the reviewer's suggestion, we have now applied multivariable fractional polynomials (by using R Package "mfp"). As the reviewer mentioned, the plots of multivariable fractional polynomials models and plots of Restricted Cubic Splines models showed very similar patterns for the dose-response and non-linearity (see below **Figure 1 for Review Only**).

We have now provided these data in the revised manuscript. In the **Methods section**: "We further used multivariable fractional polynomials to model the potential nonlinear association between serum retinol and overall and cause-specific mortality (by using R Package "mfp")." (page 12, paragraph 2, line 281-283)

In the **Results section**: "The multivariable fractional polynomials analysis, after adjustment for confounding factors, showed similar patterns of the dose-response associations between serum retinol and overall mortality as well as CVD and heart disease mortality (P values < 0.0001 for multivariable fractional polynomials compared with linear, **Supplemental Figure 1**). By contrast, serum retinol appeared to be linearly associated with respiratory disease mortality (on the log scale of the hazard ratio) (restricted cubic splines model, P nonlinear = 0.04, **Figure 1**; multivariable fractional polynomials model, P nonlinear > 0.05, **Supplemental Figure 1**)." (page 5, paragraph 1, lines 95-101)

Figure 1 for Review Only. Multivariable fractional polynomial models for the nonlinear association between serum retinol and overall and cause-specific death in the ATBC Study. The reference value (483 µg/L; hazard ratio = 1) corresponds to the cutoff value of the first quintile category of serum retinol concentration. A) Overall mortality. B) Cardiovascular disease (CVD) mortality. C) Heart disease mortality. D) Respiratory disease mortality. Multivariable fractional polynomial models adjusted for age, BMI, serum total and serum HDL cholesterol, cigarettes smoked per day, years of smoking, alcohol intake, intervention assignment, systolic and diastolic blood pressure, history of CVD, and history of diabetes.

- There are some in my view unexpected correlations between the baseline covariables, as seen in table 1. This has been mentioned (1 53-55) however is not further discussed.

Response 3: Thank you for this comment. Although the correlations observed between serum retinol and several baseline characteristics in Table 1 were formally statistically significant, the magnitude of these correlations are relatively small or negligible. According to Schober et al.,¹ when applying the conventional approach to interpreting the correlation coefficient (range from -1 to 1), it is suggested as a negligible correlation with r of 0.00 to 0.10, a weak correlation with r of 0.10 to 0.39, a moderate correlation with r of 0.40 to 0.69, a strong correlation with r of 0.70 to 0.89, and a very strong correlation with r of 0.90 to 1.00. In our data, all factors are negligibly or weakly associated with serum retinol ($0 < r \leq 0.22$, see below **Table 1 for Review Only**). We have now added this information to the Result section: “With regard to the magnitude of the correlation, serum retinol was negligibly or weakly correlated with these factors (all Pearson correlation coefficients, $-0.12 < r \leq 0.22$).¹” (page 4, paragraph 1, lines 71-73)

Table 1 for Review Only. Pearson Correlation Coefficients Between Serum Retinol and Baseline Characteristics

	Correlation coefficient with	P value
--	------------------------------	---------

	serum retinol ($\mu\text{g/L}$)	
Age	-0.12	<0.0001
Systolic blood pressure (mm Hg)	0.09	<0.0001
Diastolic blood pressure (mm Hg)	0.15	<0.0001
Serum total cholesterol (mmol/L)	0.22	<0.0001
Serum HDL cholesterol (mmol/L)	0.08	<0.0001
BMI (kg/m^2)	0.12	<0.0001
Education (% , > elementary school)	0.08	<0.0001
Physically active (%)	0.03	<0.0001
History of CVD (%) ^b	0.05	<0.0001
Vitamin A supplement use (%)	0.05	<0.0001
Vitamin E supplement use (%)	0.05	<0.0001
Alcohol (g ethanol)	0.20	<0.0001

Since for some disease groups there are major differences between the age-adjusted and the multivariable model (CVD, heart disease), further investigations of interaction effects could be useful and interesting.

Response 4: Thank you. To allay the concern of potential confounding effects, we have carefully controlled for a wide range of known potential confounding factors in the final analytical models. The non-association of serum retinol with risk of CVD- and heart disease-mortality in the **age-adjusted models**, can be explained by confounding from CVD related factors (i.e., history of CVD and systolic and diastolic blood pressure). The CVD related factors that are both associated with serum retinol status and CVD related mortality, therefore serve as the potential confounders, and should be included in the final multivariable-adjusted model (see below of **Table 2 for Review Only**). In terms of the tests for interaction, the associations of serum retinol-CVD mortality and serum retinol-heart disease mortality were not significantly modified by either CVD history or systolic and diastolic blood pressure (see below of **Table 3 for Review Only**). We have now added this information to the **Supplemental Tables 1 and 2**.

Table 2 for Review Only. Hazard ratios (HRs) and 95% CI for overall and cause-specific mortality by quintile of serum retinol

Cause of mortality	Serum retinol					P for trend ^a
	Quintile 1	Quintile 2	Quintile 3	Quintile 4	Quintile 5	
CVD						
HR (95% CI) ^b	1.00	0.86 (0.81, 0.91)	0.89 (0.84, 0.95)	0.85 (0.80, 0.90)	0.97 (0.91, 1.03)	0.50
HR (95% CI) ^c	1.00	0.85 (0.80, 0.90)	0.89 (0.84, 0.95)	0.83 (0.78, 0.88)	0.92 (0.86, 0.98)	0.023
HR (95% CI) ^d	1.00	0.84 (0.78, 0.89)	0.86 (0.81, 0.91)	0.80 (0.75, 0.85)	0.87 (0.81, 0.92)	<0.0001
Heart disease						
HR (95% CI) ^b	1.00	0.86 (0.80, 0.92)	0.90 (0.84, 0.97)	0.85 (0.79, 0.91)	0.96 (0.90, 1.03)	0.46

HR (95% CI) ^c	1.00	0.85 (0.79, 0.91)	0.90 (0.84, 0.96)	0.83 (0.77, 0.89)	0.91 (0.85, 0.98)	0.023
HR (95% CI) ^d	1.00	0.84 (0.78, 0.90)	0.87 (0.81, 0.93)	0.80 (0.75, 0.86)	0.87 (0.81, 0.93)	0.0003

^a *P* value for trend: based on statistical significance of the coefficient of the quintile variable (median value within each quintile).

^b Adjusted for age.

^c Adjusted for age, history of CVD.

^d Adjusted for age, systolic and diastolic blood pressure.

Table 3 for Review Only.

Supplemental Table 1. Hazard ratios (HRs) and 95% CI for CVD mortality by quintile of serum retinol, stratified by selected factors

	HR (95% CI) ^a	P for interaction ^b
CVD mortality		
History of CVD at baseline		
No	0.93 (0.90, 0.96)	0.24
Yes	0.98 (0.95, 1.01)	
Systolic and diastolic blood pressure		
<130	0.93 (0.88, 0.98)	0.94
130-<150	0.94 (0.91, 0.97)	
≥150	0.95 (0.92, 0.98)	
Diastolic blood pressure		
<80	0.92 (0.87, 0.98)	0.21
80-<90	0.91 (0.88, 0.95)	
≥90	0.96 (0.94, 0.99)	
Heart disease mortality		
History of CVD at baseline		
No	0.90 (0.87, 0.94)	0.27
Yes	0.98 (0.95, 1.01)	
Systolic and diastolic blood pressure		
<130	0.90 (0.85, 0.95)	0.46
130-<150	0.93 (0.90, 0.97)	
≥150	0.96 (0.93, 1.00)	
Diastolic blood pressure		
<80	0.92 (0.86, 0.98)	0.09
80-<90	0.90 (0.86, 0.94)	
≥90	0.97 (0.93, 1.00)	

Abbreviations: BMI= body mass index; CI= confidence interval; CVD= cardiovascular disease; HDL= high-density lipoprotein

^a Adjusted for age, BMI, serum total and serum HDL cholesterol, cigarettes smoked per day, years of smoking, alcohol intake, intervention assignment, systolic and diastolic blood pressure, history of CVD, and history of diabetes.

^b *P* value for interaction: according to the likelihood test to assess the statistical significance of the cross-product term entered to the Cox proportional hazard regression model.

- The selection procedure for the covariables in the multivariable model is not mentioned. I presume they have been selected *a priori* and no modelling procedure such as backward selection or a more advanced method was applied. Please discuss.

Response 5: Thank you for pointing this out. The Reviewer is correct that *a priori* covariables in the models were selected based current knowledge and prior research of potential confounding factors of serum retinol and risk of mortality (and not based on the selection modeling procedures). We have now added this information to the Methods section: “A set of *a priori* covariables were chosen for the risk models based on the existing knowledge and research regarding potential confounding of the serum retinol-mortality risk associations”. (page 12, paragraph 1, lines 266-267)

- I find the notation "1-SD" a bit strange (better: "one SD" ?) and I wonder if it is not more illustrative to report the estimated ARD for a fixed value, say 100 μ g/l.

Response 6: Thank you for suggesting this. We now use “one SD” throughout the manuscript, and labelled “130 μ g/L” as a fixed value as suggested (page 4, paragraph 2, lines 85 and 87; page 6, paragraph 2, line 128; page 12, paragraph 1, lines 263 and 274; page 13, paragraph 1, line 295; Table 2, page 21).

- the Kaplan-Meier estimates are illustrative and should not be hidden in the supplement, if possible.

Response 7: Thank you for this suggestion - we have now included the Kaplan-Meier plots as a main **Figure 2** in the manuscript. (**Figure 2**; page 5, paragraph 1, line 103)

Reviewer #2 (Remarks to the Author):

J Huang et al. present a large prospective epidemiological study on the association between serum retinol concentration and the risk of all-cause and cause-specific death among participants from the ATBC study followed for up to 30y. This is an interesting study about a vitamin which health impact has been neglected for too long, while it could be explained by its highly regulated metabolism (well-known homeostasis, which may induce a very small variability and then no significant results and publication bias). The study has indisputable strengths, including the original topic, the large sample size, the prospective design.

Response 8: We appreciate the Reviewer’s positive overall assessment. We have addressed each of their specific comments below, with the manuscript modified accordingly.

It is generally acknowledged that the circulating retinol concentration is highly regulated, to ensure a secure healthy context. Therefore, exhibiting very low levels of circulating retinol would reflect a potential liver storage issue, while excess of retinol levels would also be accompanied by adverse health outcomes. It’s surprising not to have discuss these points here, and not to have define an upper limit not to surpass. However, results did not support such hypothesis.

Response 9: Thank you for this comment. We agree with the Reviewer that serum retinol levels are under homeostatic control,² and that individual differences in serum retinol concentrations may reflect variability in liver storage and function, genetic predisposition, as well as dietary intake.^{2, 3, 4} The optimal range of serum retinol, rather than insufficient or excess concentrations, can be assumed to be associated with beneficial long-term health outcomes. In our data, we do find the optimal levels of serum retinol (600-700 μ g/L) were associated with a lower risk of overall and cause-specific mortality. However, our findings do not support the hypothesis of a remarkable relationship between excessive concentrations of serum retinol and increased risk of mortality, which may be due to the range values of the serum retinol concentration in our study population.

Whether there is a positive association of mortality risk with excessive concentrations of serum retinol (e.g., >1200 µg/L) can be evaluated in other study populations with very high serum retinol values.

We have therefore modified the Discussion section to include this issue. “Whole body storage of vitamin A is highly homeostatically regulated, and individual differences in concentrations may reflect variability in liver storage and function, genetic predisposition as well as dietary intake.^{2, 3, 4} Previous studies raised the possibility that both insufficient and excess vitamin A status may be associated with increased risk of adverse health outcomes,^{5, 6} including as a result of direct toxic properties of hypervitaminosis A.² We did not observe excess mortality among men with high serum retinol, however, possibly because of a limited range of concentrations in our study population. The threshold of toxic concentrations of retinol, as well as whether there is a positive association of mortality risk with excess concentrations of serum retinol, can be evaluated in further studies of populations with available data of excessive serum retinol values.” (on the bottom of page 6, lines 135-136; on the top of page 7, lines 137-144)

Abstract

Authors should privilege the most interesting results, also being enough informative. However, here there is no number describing higher risk of death (how many?), no CI 95%. This is important since the sample size may help finding statistically significant results, with low clinical relevance. What are the subgroups? What are the number of cause-specific deaths?

Response 10: Thank you for pointing this out. We have now provided the suggested information in the Abstract. “During a 30-year cohort follow-up through Finnish registry linkage, there were 23,797 deaths, including from cardiovascular disease (9,869; 8,064 from heart disease and 1,764 from stroke), cancer (7,695), respiratory disease (2,161), diabetes (119), injuries and accidents (1,255), and other causes (2,698). Participants with higher serum retinol experienced significantly lower overall, CVD, heart disease, and respiratory disease mortality compared to men with the lowest retinol concentrations, reflecting 17% to 32% lower mortality risk (overall mortality, multivariable-adjusted HRs (95% CIs)=0.87 (0.83, 0.90), 0.84 (0.80, 0.87), 0.80 (0.77, 0.83,) and 0.83 (0.80, 0.87) for quintile 2 [Q2]-Q5 versus Q1, respectively; $P_{\text{trend}} < 0.0001$). The retinol-overall mortality association was similar across several cohort subgroups, including smoking intensity, alcohol consumption, body mass index, trial supplementation, serum alpha-tocopherol and beta-carotene concentrations, and duration of follow-up. Mediation analysis indicated that <3% of the effects of duration of smoking and diabetes mellitus on mortality were mediated through retinol concentration.” (Abstract; page 2, paragraph 1, lines 26-36)

Introduction

This is a short and well-written introduction, which could be more informative about limitations of previous studies (it is mentioned that sample size are limited, such as follow-up durations, confounding factors (which ones?)) and results from diverse sub-sample groups (which ones?) should be underlined.

Response 11: We appreciate positive and encouraging evaluation. We have now added more information to the introduction. “However, most of the previous studies were of limited sample size and number of events (i.e., ranging from 62 to 720 deaths),^{7, 8, 9, 10, 11} controlled for confounding factors inconsistently (e.g., only controlling for age and sex⁹), and had relatively low power to

examine cause-specific mortality, effect modification by other factors,^{5, 7, 8, 9, 10, 12} and dose-response associations.^{7, 8, 9, 10,} (page 3, paragraph 2, lines 55-58)

Methods:

This section is well-written and comprehensively described. It remains some minor points that should be addressed:

Response 12: Thank you for the positive overall evaluation.

- Why duration and intensity of smoking were both considered as adjustment variables? It appears these variables could be co-linear and may be not both useful in the same model

Response 13: Both duration and intensity of smoking reflect different dimensions of smoking history and status that *independently* impact mortality. In addition, the correlation between duration and intensity of smoking (Pearson correlation coefficient=0.067) would be considered as small correlation based on a report from Schober et al. (i.e., Pearson correlation coefficient, $0 \leq r < 0.1$, negligible correlation).¹ In order to more thoroughly control for smoking and avoid potential residual confounding, our final models included both duration and intensity.

- Several stratified analyses have been performed, while it should better justify.

Response 14: We have now included the following justification: “In order to evaluate potential interactions and the robustness of our findings, stratified analyses were constructed based on *a priori* categories of baseline age (<54, 54-<59, or >59 years), daily cigarettes (<16, 16-20, or >20 cigarettes per day), alcohol consumption...” in the Methods section. (page 13, paragraph 1, lines 286-287)

- The risk of death after 10, 20 and 30y of follow-up has been reported. Why? Did the authors justify why they considered these thresholds?

Response 15: Thank you for the comment. The truncation of follow-up time is arbitrary with prespecified length, and the aim of this analysis was to examine whether the risk estimates remained qualitatively similar at different truncating follow-up time. We have now included this information in the Results section: “The findings of absolute risk difference for the associations remained qualitatively similar when truncating the follow-up observation periods at 10- or 20- years”. (page 4, paragraph 2, lines 85-87)

- Results

- Table 1. Please include at the top of each column the pertinent data. Either the median, the mean, or the range of serum retinol concentrations.

Response 16: We have now included the mean and standard deviation values of the serum retinol concentrations at the top of the quintile columns in Table 1. (page 20)

- Bonferroni: are all results presented after correction for multiple comparisons ?

Response 17: Yes, we have previously mentioned that the predefined Bonferroni correction threshold was used. “The Bonferroni correction threshold was applied to control for multiple comparisons ($0.05/9=0.0056$ for primary and secondary analyses [nine tests], and $0.05/10=0.005$ for

ten effect modification tests across the subgroups).” (page 14, paragraph 1, lines 312-314)

As the Reviewer pointed it out, we have now provided the Bonferroni corrected P-value in **Table 2**. (pages 21-22)

Discussion:

- The homeostasis of serum retinol concentrations should be acknowledged as a limit. This means that the small variation of the serum retinol concentration between participants of this sample might rather reflect some particular dietary habits. There are some references about the circulating retinol concentration, which appears associated with dietary intakes (offal such as liver). Adding some data about the dietary intakes of participants would have improve the interpretation of data.

Response 18: Thank you for this comment, and we agree that diet along with genetic predisposition would account for variation in retinol status. We have now provided information to discuss this issue. “Whole body storage of vitamin A is highly homeostatically regulated, and individual differences in concentrations may reflect variability in liver storage and function, genetic predisposition as well as dietary intake.^{2, 3, 4}” (on the bottom of page 6, lines 135-136 and on the top of page 7, line 137) and “Given the fact that vitamin A is metabolized and stored in the liver,² a positive association between liver consumption and serum retinol is biologically plausible. However, along with previous findings,^{3, 13, 14, 15, 16} our data showed a negligible association between serum retinol and dietary liver consumption (Pearson correlation coefficient=0.05, P<0.0001), and a weak association with alcohol consumption (Pearson correlation coefficient=0.20, P<0.0001), with negligible and weak correlations being defined as correlation coefficients that range from 0.00-0.10 and 0.10-0.39, respectively.¹ Additionally, our data showed that the inverse association between serum retinol and risk of overall mortality was not changed by additional adjustment for liver consumption (Q5 versus Q1: HR=0.84, 95% CI: 0.80, 0.87, P for trend<0.0001; data not shown). This suggests that a vitamin A-enriched diet was not sufficient to modify the associations appreciably.” (on the top of page 7, lines 144-153)

- A single retinol assessment is available; some relevant markers of the vitamin A status (RBP, retinoic acid, retinoic acid receptor expression) would have reinforced the results, while it’s easily understandable that such measures were not available for a so large sample. Overall, it remains minor issues that should be addressed to improve this paper.

Catherine Féart

Response 19: We agree with the reviewer that data for the additional vitamin A related markers would have been very interesting to examine along with retinol, but those measures were not available for the large cohort. We have now put this as a limitation in the Discussion section. “Other biomarkers relevant to vitamin A status, including retinol-binding protein and retinoic acid receptor expression, that would have afforded a deeper evaluation of the mortality associations, were not available for the cohort.” (on the top of page 10, lines 212-214)

References

1. Schober P, Boer C, Schwarte LA. Correlation Coefficients: Appropriate Use and Interpretation. *Anesthesia & Analgesia* **126**, 1763-1768 (2018).
2. Tanumihardjo SA, *et al.* Biomarkers of Nutrition for Development (BOND)-Vitamin A Review. *J Nutr*

- 146**, 1816S-1848S (2016).
3. Fearf C, *et al.* Plasma retinol and association with socio-demographic and dietary characteristics of free-living older persons: the Bordeaux sample of the three-city study. *Int J Vitam Nutr Res* **80**, 32-44 (2010).
 4. Mondul AM, *et al.* Genome-wide association study of circulating retinol levels. *Hum Mol Genet* **20**, 4724-4731 (2011).
 5. Min KB, Min JY. Relation of serum vitamin A levels to all-cause and cause-specific mortality among older adults in the NHANES III population. *Nutr Metab Cardiovasc Dis* **24**, 1197-1203 (2014).
 6. Michaelsson K, Lithell H, Vessby B, Melhus H. Serum retinol levels and the risk of fracture. *N Engl J Med* **348**, 287-294 (2003).
 7. Fletcher AE, Breeze E, Shetty PS. Antioxidant vitamins and mortality in older persons: findings from the nutrition add-on study to the Medical Research Council Trial of Assessment and Management of Older People in the Community. *Am J Clin Nutr* **78**, 999-1010 (2003).
 8. Ito Y, *et al.* A population-based follow-up study on mortality from cancer or cardiovascular disease and serum carotenoids, retinol and tocopherols in Japanese inhabitants. *Asian Pac J Cancer Prev* **7**, 533-546 (2006).
 9. Bates CJ, Hamer M, Mishra GD. Redox-modulatory vitamins and minerals that prospectively predict mortality in older British people: the National Diet and Nutrition Survey of people aged 65 years and over. *Br J Nutr* **105**, 123-132 (2011).
 10. Connolly GM, Cunningham R, Maxwell AP, Young IS. Decreased serum retinol is associated with increased mortality in renal transplant recipients. *Clin Chem* **53**, 1841-1846 (2007).
 11. Liu Y, *et al.* Association of Serum Retinoic Acid With Risk of Mortality in Patients With Coronary Artery Disease. *Circ Res* **119**, 557-563 (2016).
 12. Goyal A, Terry MB, Siegel AB. Serum antioxidant nutrients, vitamin A, and mortality in U.S. Adults. *Cancer Epidemiol Biomarkers Prev* **22**, 2202-2211 (2013).
 13. Yu Y, *et al.* Plasma retinol and the risk of first stroke in hypertensive adults: a nested case-control study. *Am J Clin Nutr* **109**, 449-456 (2019).
 14. Gey KF, *et al.* Low plasma retinol predicts coronary events in healthy middle-aged men: the PRIME Study. *Atherosclerosis* **208**, 270-274 (2010).
 15. Moran NE, Mohn ES, Hason N, Erdman JW, Jr., Johnson EJ. Intrinsic and Extrinsic Factors Impacting Absorption, Metabolism, and Health Effects of Dietary Carotenoids. *Adv Nutr* **9**, 465-492 (2018).
 16. Woodside JV, *et al.* Factors associated with serum/plasma concentrations of vitamins A, C, E and carotenoids in older people throughout Europe: the EUREYE study. *Eur J Nutr* **52**, 1493-1501 (2013).
-
-

Please let us know if you have any additional questions or suggestions. We can be most-easily reached at JIAQI.HUANG@LIVE.COM or DAA@NIH.GOV

Sincerely yours,

Jiaqi Huang, Demetrius Albanes

Jiaqi Huang, Ph.D., M.S.

Demetrius Albanes, M.D.

Metabolic Epidemiology Branch

Division of Cancer Epidemiology and Genetics

Reviewers' Comments:

Reviewer #1:

Remarks to the Author:

Thank you very much for the careful revision and for the detailed answers to my questions

Reviewer #3:

Remarks to the Author:

I write to provide my review of this very interesting and well written manuscript entitled "Association Between Serum Retinol and Overall and Cause-Specific Mortality: A 30-Year Prospective Cohort Study." I understand that this manuscript has been reviewed before at Nature Communications and that whilst I did not see the first version of the paper, I am requested to comment on this revised version and assess the authors' response to Reviewer 2, who was unavailable to comment on the revised version. I further understand that at this stage you do not wish to generate additional major revisions for the authors, however if I note any fundamental issues that I am to let you know. At this advice, I am providing the requested information. Reviewer 2 indicated that, "It is generally acknowledged that the circulating retinol concentration is highly regulated..." In response, the authors have stated in multiple locations that, "Whole body storage of vitamin A is highly homeostatically regulated, and individual differences in concentrations may reflect variability in liver storage and function, genetic predisposition as well as dietary intake." I believe that the authors have misunderstood reviewer 2's comment, which specifically refers to circulating retinol concentrations. The tissue stores, and particularly liver stores, are likely highly variable depending on dietary intake. But release of retinol into the circulation is dependent on the release of retinol bound to retinol binding protein which then circulate at a one-to-one stoichiometry. The release of holo RBP from the liver is highly regulated through various mechanisms and for this reason, serum retinol is not thought to be a very good indicator of liver retinoid stores except under conditions of vitamin A deficiency. In vitamin A deficiency, liver retinol levels are low and hepatic RBP accumulates, so serum concentrations of both retinol and RBP decline. This brings up several related points:

- This aspect of retinoid physiology should be discussed at least briefly.
- It is unlikely that this relatively well-nourished study population has a high degree of low liver retinol stores. It would be useful to know how many study participants met WHO criteria for severe, mild, or moderate vitamin A deficiency according to serum retinol cut-offs. This data could be added to Table 1 and mentioned in the discussion.
- One of the factors that influences release of RBP bound retinol from the liver is inflammation. RBP is a negative acute phase protein, so during inflammation (even mild inflammation due to subclinical disease), serum retinol and RBP levels are reduced. It would be helpful if the authors had a marker of inflammation (e.g. CRP) to control for this potential source of confounding. If they do not have this data, it should be listed as a potential confounder and mentioned in the Discussion. In the NHANES III study linking serum retinol to all-cause and cause-specific mortality, models were adjusted for CRP.

Minor comments:

- The authors added data on the association between serum retinol and dietary liver consumption (pg 7) showing a weak Pearson correlation coefficient. Given that the data are available, it would be useful to know how common / rare liver consumption is in the study population, and if rare, what are the primary sources of dietary vitamin A in this population.
- Given that the authors mention availability of data on serum retinol at year 3, it would be interesting to see if the authors considered doing a sensitivity analysis using year 3 data.
- In the Discussion the authors might refer to vitamin A as plasma (or serum) retinol throughout for consistency.

National Institutes of Health
National Cancer Institute
Bethesda, Maryland 20892
NCI-Shady Grove – 6E316

RE: NCOMMS-20-46725A: Association Between Serum Retinol and Overall and Cause-Specific Mortality: A 30-Year Prospective Cohort Study

Reviewer Comments:

Reviewer #1 (Remarks to the Author):

Thank you very much for the careful revision and for the detailed answers to my questions

Response 1: We greatly appreciated your previous comments and the positive overall evaluation.

Reviewer #3 (Remarks to the Author):

I write to provide my review of this very interesting and well written manuscript entitled "Association Between Serum Retinol and Overall and Cause-Specific Mortality: A 30-Year Prospective Cohort Study." I understand that this manuscript has been reviewed before at Nature Communications and that whilst I did not see the first version of the paper, I am requested to comment on this revised version and assess the authors' response to Reviewer 2, who was unavailable to comment on the revised version. I further understand that at this stage you do not wish to generate additional major revisions for the authors, however if I note any fundamental issues that I am to let you know. At this advice, I am providing the requested information.

Response 2: We greatly appreciated your effort to complete the review process of our manuscript based on our response to Reviewer #2, and we thank you for the favorable overall assessment. We have now addressed each of your comments below, with the manuscript modified accordingly.

Reviewer 2 indicated that, "It is generally acknowledged that the circulating retinol concentration is highly regulated..." In response, the authors have stated in multiple locations that, "Whole body storage of vitamin A is highly homeostatically regulated, and individual differences in concentrations may reflect variability in liver storage and function, genetic predisposition as well as dietary intake." I believe that the authors have misunderstood reviewer 2's comment, which specifically refers to circulating retinol concentrations. The tissue stores, and particularly liver stores, are likely highly variable depending on dietary intake. But release of retinol into the circulation is dependent on the release of retinol bound to retinol binding protein which then circulate at a one-to-one stoichiometry. The release of holo RBP from the liver is highly regulated through various mechanisms and for this reason, serum retinol is not thought to be a very good indicator of liver retinoid stores except under conditions of vitamin A deficiency. In vitamin A deficiency, liver retinol levels are low and hepatic RBP accumulates, so serum concentrations of both retinol and RBP decline. This brings up several related points:

- This aspect of retinoid physiology should be discussed at least briefly.

Response 3: Thank you for the clarification of the comment of Reviewer 2. We have now provided greater detail regarding vitamin A physiology and homeostasis. In the **Discussion section**: “Humans cannot synthesize vitamin A and derive this essential nutrient from animal product-based retinyl esters and plant-based pro-vitamin A carotenoids (e.g., alpha- and beta-carotene).^{1, 2}” and “Preformed retinol is absorbed from the intestine, esterified, and transported in chylomicrons to the liver. For healthy, well-nourished individuals, it is estimated that 60-95% of vitamin A is stored in the liver which plays a central role in its metabolism and homeostasis through hydrolysis of stored retinyl esters, complexing of retinol with retinol-binding protein (RBP), and release of this “holo-RBP” into systemic circulation for uptake and use by other organs.^{1, 2, 3, 4}” (on the top of page 7, lines 141-143 and lines 145-150)

- It is unlikely that this relatively well-nourished study population has a high degree of low liver retinol stores. It would be useful to know how many study participants met WHO criteria for severe, mild, or moderate vitamin A deficiency according to serum retinol cut-offs. This data could be added to Table 1 and mentioned in the discussion.

Response 4: Thank you for this valuable suggestion, and you are correct that we do not observe evidence of low liver stores. According to the WHO criteria, a serum retinol concentration of ≤ 98 $\mu\text{g/L}$ indicates severe vitamin A deficiency, and a concentration of 98 to ≤ 196 $\mu\text{g/L}$ indicates subclinical vitamin A deficiency. In the present ATBC Study population, we only have two participants ($n=2$) identified as severe vitamin A deficiency, and 21 participants identified as subclinical vitamin A deficiency. We now provide this information in the **Results section**: “According to the WHO criteria, the present study had only two participants identified with severe vitamin A deficiency (≤ 98 $\mu\text{g/L}$), and 21 participants identified with subclinical vitamin A deficiency (98 to ≤ 196 $\mu\text{g/L}$)”. (page 4, paragraph 1, lines 67-69) In the **Discussion section**: “Our data suggest mortality declines with increasing retinol concentration, with a risk nadir for overall and CVD mortality in older men with serum values of 600-700 $\mu\text{g/L}$, and excess mortality for men with retinol below 500 $\mu\text{g/L}$ ($n=7,321$, which included two and 21 participants that exhibited severe [≤ 98 $\mu\text{g/L}$] or subclinical [98 to ≤ 196 $\mu\text{g/L}$] vitamin A deficiency, respectively).” (on the top of page 7, lines 139-141)

- One of the factors that influences release of RBP bound retinol from the liver is inflammation. RBP is a negative acute phase protein, so during inflammation (even mild inflammation due to subclinical disease), serum retinol and RBP levels are reduced. It would be helpful if the authors had a marker of inflammation (e.g. CRP) to control for this potential source of confounding. If they do not have this data, it should be listed as a potential confounder and mentioned in the Discussion. In the NHANES III study linking serum retinol to all-cause and cause-specific mortality, models were adjusted for CRP.

Response 5: Thank you for this comment. The ATBC Study does not contain the serum CRP concentration data, therefore we cannot include this potential confounder in our models. We have now included this in the limitations of the Discussion: “Other biomarkers relevant to vitamin A status, including retinol-binding protein, retinoic acid receptor expression and **C-reactive protein (CRP, a marker of inflammation, that the inflammation status may affect retinol homeostasis and serum retinol concentrations)**, that would have afforded a deeper evaluation of the mortality associations, were not available for the cohort.” (page 10, paragraph 1, lines 224-228)

Minor comments:

- The authors added data on the association between serum retinol and dietary liver consumption (pg 7) showing a weak Pearson correlation coefficient. Given that the data are available, it would be useful to know how common / rare liver consumption is in the study population, and if rare, what are the primary sources of dietary vitamin A in this population.

Response 6: Median liver consumption in our cohort was of 3.6 g/d (interquartile range = 0.41 to 7.5 g/d), which might indicate consuming a 100 g portion once a month. The primary dietary sources for dietary vitamin A in this population (as mean percentage of total daily dietary vitamin A) were liver (33.9%), butter (16.4%), egg (9.2%), milk (8.6%), cheese (3.5%) and other food items combined (28.4%).

We have now provided this information in the **Discussion section**: “For example, the primary dietary sources for dietary vitamin A in this population (as mean percentage of total daily dietary vitamin A) were liver (33.9%), butter (16.4%), egg (9.2%), milk (8.6%), cheese (3.5%) and other food items combined (28.4%).” (on the top of page 7, lines 143-145)

- Given that the authors mention availability of data on serum retinol at year 3, it would be interesting to see if the authors considered doing a sensitivity analysis using year 3 data.

Response 7: Thank you for this comment. Based on the reviewer’s suggestion, we have performed this sensitivity analysis using serum retinol data from year 3 and found the associations with mortality were essentially the same as our original findings that used baseline serum concentrations. We have now added this secondary analysis to the manuscript. In the **Methods** section: “In a sensitivity analysis, we examined serum retinol concentrations measured from blood collected in year three in relation to subsequent overall and cause-specific mortality (n= 22,312 men included). Baseline and year 3 serum retinol were highly correlated (Spearman correlation coefficient 0.69.” (on the top of page 14, lines 313-316) In the **Results** section: “The findings were similar after excluding the first 5 years of follow-up (**Supplemental Table 5**) and when used serum retinol concentration data from the third follow-up year (fifth versus first quintile, HR [95% CIs] for overall mortality: 0.85 (0.81, 0.89), $P_{\text{trend}} < 0.0001$; **Supplemental Table 6**)”. (on the top of page 6, lines 118-120)

Supplemental Table 6. Multivariate-adjusted hazard ratios (HRs) for overall and cause-specific mortality by quintile of serum retinol concentration at three years in the ATBC Study ^a

Causes of mortality	Serum retinol (mg/L)						P for trend	Bonferroni corrected P for trend
	Quintile 1	Quintile 2	Quintile 3	Quintile 4	Quintile 5			
All-cause								

Deaths (n)	3180	3415	3301	3578	4196		
Age-adjusted HR (95% CI) ^b	1.00	0.89 (0.84, 0.93)	0.85 (0.81, 0.89)	0.84 (0.80, 0.88)	0.90 (0.86, 0.94)	0.0002	0.0018
Multivariate HR (95% CI) ^c	1.00	0.88 (0.84, 0.93)	0.84 (0.80, 0.89)	0.82 (0.78, 0.86)	0.85 (0.81, 0.89)	<0.0001	<0.0001
CVD							
Deaths (n)	1209	1377	1359	1473	1769		
Age-adjusted HR (95% CI) ^b	1.00	0.95 (0.88, 1.02)	0.93 (0.86, 1.01)	0.92 (0.86, 1.00)	1.01 (0.94, 1.09)	0.47	1.00
Multivariate HR (95% CI) ^c	1.00	0.91 (0.85, 0.99)	0.89 (0.82, 0.96)	0.85 (0.79, 0.92)	0.88 (0.82, 0.95)	0.0016	0.01
Heart disease							
Deaths (n)	1000	1114	1112	1202	1421		
Age-adjusted HR (95% CI) ^b	1.00	0.93 (0.85, 1.01)	0.92 (0.85, 1.01)	0.91 (0.84, 0.99)	0.98 (0.91, 1.07)	0.99	1.00
Multivariate HR (95% CI) ^c	1.00	0.89 (0.82, 0.97)	0.87 (0.80, 0.95)	0.83 (0.76, 0.90)	0.84 (0.77, 0.91)	0.0001	0.0009
Stroke							
Deaths (n)	200	258	244	264	338		
Age-adjusted HR (95% CI) ^b	1.00	1.07 (0.89, 1.29)	1.01 (0.84, 1.22)	1.00 (0.83, 1.21)	1.18 (0.99, 1.40)	0.091	0.82
Multivariate HR (95% CI) ^c	1.00	1.07 (0.89, 1.29)	1.02 (0.84, 1.23)	0.99 (0.82, 1.19)	1.12 (0.93, 1.34)	0.35	1.00
Cancer							
Deaths (n)	1029	1141	1088	1201	1371		
Age-adjusted HR (95% CI) ^b	1.00	0.91 (0.84, 0.99)	0.87 (0.80, 0.94)	0.87 (0.80, 0.95)	0.90 (0.83, 0.98)	0.024	0.22
Multivariate HR (95% CI) ^c	1.00	0.92 (0.84, 1.00)	0.88 (0.80, 0.95)	0.87 (0.80, 0.95)	0.90 (0.82, 0.97)	0.019	0.17
Respiratory disease							
Deaths (n)	391	342	316	309	307		
Age-adjusted HR (95% CI) ^b	1.00	0.73 (0.63, 0.84)	0.68 (0.58, 0.78)	0.61 (0.52, 0.71)	0.56 (0.48, 0.65)	<0.0001	<0.0001
Multivariate HR (95% CI) ^c	1.00	0.77 (0.67, 0.90)	0.74 (0.64, 0.86)	0.67 (0.57, 0.78)	0.62 (0.53, 0.72)	<0.0001	<0.0001
Diabetes mellitus							
Deaths (n)	16	17	16	19	28		
Age-adjusted HR (95% CI) ^b	1.00	0.87 (0.44, 1.73)	0.82 (0.41, 1.64)	0.89 (0.46, 1.74)	1.20 (0.65, 2.23)	0.73	1.00
Multivariate HR (95% CI) ^c	1.00	0.89 (0.45, 1.77)	0.72 (0.36, 1.45)	0.80 (0.41, 1.56)	0.96 (0.51, 1.82)	0.97	1.00
Injuries and accidents							
Deaths (n)	129	166	130	164	236		
Age-adjusted HR (95% CI) ^b	1.00	1.05 (0.83, 1.32)	0.81 (0.63, 1.03)	0.92 (0.73, 1.16)	1.18 (0.95, 1.46)	0.10	0.90
Multivariate HR (95% CI) ^c	1.00	1.08 (0.85, 1.35)	0.82 (0.64, 1.05)	0.92 (0.73, 1.16)	1.12 (0.89, 1.40)	0.38	1.00

Other causes							
Deaths (n)	406	372	392	412	485		
Age-adjusted HR (95% CI) ^b	1.00	0.73 (0.64, 0.85)	0.76 (0.66, 0.87)	0.73 (0.64, 0.84)	0.78 (0.68, 0.89)	0.007	0.06
Multivariate HR (95% CI) ^c	1.00	0.73 (0.64, 0.85)	0.76 (0.66, 0.87)	0.72 (0.62, 0.82)	0.74 (0.64, 0.84)	0.0004	0.004

Abbreviations: ATBC=Alpha-Tocopherol, Beta-Carotene Cancer Prevention; BMI= body mass index; CI= confidence interval; CVD= cardiovascular disease; HDL= high-density lipoprotein

^a There were 22,312 men included in this sensitivity analysis.

^b Adjusted for age. *P* value for trend: based on statistical significance of the coefficient of the quintile variable (median value within each quintile).

^c Adjusted for age, BMI, serum total and serum HDL cholesterol, cigarettes smoked per day, years of smoking, alcohol intake, intervention assignment, systolic and diastolic blood pressure, history of CVD, and history of diabetes.

- In the Discussion the authors might refer to vitamin A as plasma (or serum) retinol throughout for consistency.

Response 8: Thank you pointing this out. We now refer to vitamin A as circulating (or serum) retinol for consistency in the Discussion. (page 6, paragraph 2, lines 129-130)

References

1. Carazo A, Macakova K, Matousova K, Krcmova LK, Protti M, Mladenka P. Vitamin A Update: Forms, Sources, Kinetics, Detection, Function, Deficiency, Therapeutic Use and Toxicity. *Nutrients* **13**, (2021).
2. Tanumihardjo SA, *et al.* Biomarkers of Nutrition for Development (BOND)-Vitamin A Review. *J Nutr* **146**, 1816S-1848S (2016).
3. Yee MMF, Chin KY, Ima-Nirwana S, Wong SK. Vitamin A and Bone Health: A Review on Current Evidence. *Molecules* **26**, (2021).
4. Borel P, Desmarchelier C. Genetic Variations Associated with Vitamin A Status and Vitamin A Bioavailability. *Nutrients* **9**, (2017).

Please let us know if you have any additional questions or suggestions. We can be most-easily reached at JIAQI.HUANG@LIVE.COM or DAA@NIH.GOV

Sincerely yours,

Jiaqi Huang, Demetrius Albanes

Jiaqi Huang, Ph.D., M.S.
Demetrius Albanes, M.D.
Metabolic Epidemiology Branch
Division of Cancer Epidemiology and Genetics

Reviewers' Comments:

Reviewer #3:

Remarks to the Author:

The authors have addressed all of my concerns and I suggest accepting this manuscript for publication.

National Institutes of Health
National Cancer Institute
Bethesda, Maryland 20892
NCI-Shady Grove – 6E316

RE: NCOMMS-20-46725C: Association Between Serum Retinol and Overall and Cause-Specific Mortality: A 30-Year Prospective Cohort Study

Reviewer Comments:

Reviewer #3 (Remarks to the Author):

The authors have addressed all of my concerns and I suggest accepting this manuscript for publication.

Response 1: We greatly appreciated your previous comments and the positive overall evaluation.

Please let us know if you have any additional questions or suggestions. We can be most-easily reached at JIAQI.HUANG@LIVE.COM or DAA@NIH.GOV

Sincerely yours,

Jiaqi Huang, Demetrius Albanes

Jiaqi Huang, Ph.D., M.S.
Demetrius Albanes, M.D.
Metabolic Epidemiology Branch
Division of Cancer Epidemiology and Genetics